# Management and Clinical Outcomes of Breast Cancer in Women Diagnosed with Hereditary Cancer Syndromes in a Clinic-Based Sample from Colombia

**DOI:** 10.3390/cancers16112020

**Published:** 2024-05-26

**Authors:** María Carolina Sanabria-Salas, Ana Pedroza-Duran, Sandra E. Díaz-Casas, Marcela Nuñez Lemus, Carlos F. Grillo-Ardila, Ximena Briceño-Morales, Mauricio García-Mora, Javier Ángel-Aristizábal, Iván Fernando Mariño Lozano, Raúl Alexis Suarez Rodríguez, Luis Hernán Guzmán Abisaab

**Affiliations:** 1Instituto Nacional de Cancerología, Calle 1 N. 9-85, Bogotá 111511, Colombia; ampedroza@fucsalud.edu.co (A.P.-D.); sdiaz@cancer.gov.co (S.E.D.-C.); mnunez@cancer.gov.co (M.N.L.); elmastocito@yahoo.com (X.B.-M.); magarcia@cancer.gov.co (M.G.-M.); jaangel@cancer.gov.co (J.Á.-A.); imarino@cancer.gov.co (I.F.M.L.); rasuarez@cancer.gov.co (R.A.S.R.); lguzmana@cancer.gov.co (L.H.G.A.); 2Division of Medical Oncology and Hematology, Princess Margaret Cancer Centre, University Health Network, 610 University Avenue, Toronto, ON M5G 2C1, Canada; 3Department of Obstetrics & Gynecology, School of Medicine, Universidad Nacional de Colombia, Avenida Carrera 30 N. 45-3, Bogotá 111321, Colombia; cfgrilloa@unal.edu.co

**Keywords:** breast neoplasms, genetic predisposition to disease, hereditary breast and ovarian cancer syndrome, bilateral risk-reducing mastectomy, prevention, survival

## Abstract

**Simple Summary:**

Hereditary breast cancer represents a significant proportion of breast cancer cases worldwide, and *BRCA1* and *BRCA2* explain the majority of these. Less frequent genes with variable penetrance remain less characterized in Colombia. This study explores the characteristics and outcomes of 82 breast cancer patients with germline pathogenic/likely pathogenic variants (PVs) treated and followed at the Instituto Nacional de Cancerología, Colombia (INC-C) from 2018 to 2021 as part of an Institutional Hereditary Cancer Program implementation. We described the distribution of germline PVs among the different breast cancer molecular subtypes in this cohort of carriers. We observed that most PVs were in known breast cancer susceptibility genes, and PVs in *BRCA2* were the most frequent. Notably, we observed that patients with disease progression were predominantly carriers of PVs in the *BRCA2* gene.

**Abstract:**

This study aimed to investigate prognosis and survival differences in 82 breast cancer patients with germline pathogenic/likely pathogenic variants (PVs) treated and followed at the Breast Unit of the Instituto Nacional de Cancerología, Colombia (INC-C) between 2018 and 2021. Median age at diagnosis was 46 years, with 62.2% presenting locally advanced tumors, 47.6% histological grade 3, and 35.4% with triple-negative breast cancer (TNBC) subtype. Most carriers, 74.4% (61/82), had PVs in known breast cancer susceptibility genes (i.e., “associated gene carriers” group, considered inherited breast cancer cases): *BRCA2* (30), *BRCA1* (14), *BARD1* (4), *RAD51D* (3), *TP53* (2), *PALB2* (2), *ATM* (2), *CHEK2* (1), *RAD51C* (1), *NF1* (1), and *PTEN* (1). *BRCA1-2* represented 53.7%, and homologous recombination DNA damage repair (HR-DDR) genes associated with breast cancer risk accounted for 15.9%. Patients with PVs in non-breast-cancer risk genes were combined in a different category (21/82; 25.6%) (i.e., “non-associated gene carriers” group, considered other breast cancer cases). Median follow-up was 38.1 months, and 24% experienced recurrence, with 90% being distant. The 5-year Disease-Free Survival (DFS) for inherited breast cancer cases was 66.5%, and for other breast cancer cases it was 88.2%. In particular, for carriers of PVs in the *BRCA2* gene, it was 37.6%. The 5-year Overall Survival (OS) rates ranged from 68.8% for those with PVs in *BRCA2* to 100% for those with PVs in other HR-DDR genes. Further studies are crucial for understanding tumor behavior and therapy response differences among Colombian breast cancer patients with germline PVs.

## 1. Introduction

Breast cancer is the most commonly diagnosed cancer in women worldwide. In 2022, there were 2,296,840 new cases reported, and it was the fourth leading cause of mortality, with 666.103 deaths. In Colombia, according to Globocan 2022 data, the incidence and mortality rates stand at 50.7 (17,018 new cases) and 13.3 per 100,000 population, respectively [1,2]. In general, 5–10% of all cases are explained by inherited germline pathogenic or likely pathogenic variants (PVs) in cancer predisposition genes, and these are known as hereditary breast cancer syndromes [3,4]. Typically, these syndromes exhibit an autosomal dominant inheritance pattern, with a 50% probability of transmission to offspring and variable penetrance; certain characteristics, such as early onset (i.e., <45 years), multiple tumors in the same individual (i.e., bilateral presentation), a positive family history (i.e., associated cancers in close relatives from the same lineage), specific breast cancer subtypes (i.e., triple-negative, TNBC) and atypical sex distribution (i.e., breast cancer in males), are important criteria outlined by international guidelines for detecting these cases [5,6].

Hereditary breast and ovarian cancer (HBOC) syndrome, caused by PVs in *BRCA1* and *BRCA2*, has been extensively studied worldwide and is known to be associated with an increased risk of breast and ovarian cancer [3,7,8,9,10,11,12,13]. Nowadays, with the introduction of multigene panel testing, it is recognized that PVs in *BRCA* genes are responsible for more than half of all hereditary breast cancer syndromes [14,15,16]. Given the implementation of these panels, other genes with high and moderate risks have been identified. According to the NCCN guidelines v2.2024 (27 September 2023), germline PVs in the genes *ATM*, *BARD1*, *RAD51C*, *RAD51D*, *NF1*, and *CHEK2* confer an absolute breast cancer risk between 17% and 40%. Meanwhile, for the genes *CDH1*, *PALB2*, and *PTEN*, the absolute risk ranges from 40% to 60% [5]. Similarly, PVs in *STK11* are associated with a breast cancer risk estimated to be between 32% and 54% [5]. *BRCA1, BRCA2,* and *TP53* genes remain positioned as having the highest risk for breast cancer (>60%) [5].

Three studies using multigene panels for germline testing including Colombian breast/ovarian cancer cases have been published, reporting a yield of PVs of 11% to 22% [15,16,17]. Despite these reports, there is insufficient characterization regarding the impact of *BRCA* and non-*BRCA* germline PVs on the survival and outcomes among the population affected by hereditary breast cancer syndromes in the Colombian context.

In this study, we aimed to describe the clinicopathological features, prognosis, and survival rates of newly diagnosed breast cancer patients managed and followed within the same institution who were identified as carriers of germline PVs through a genetic cancer risk assessment program. Moreover, we provide information on the distribution of these genetic findings across different molecular subtypes, contributing to the characterization of the Colombian breast cancer patient population.

## 2. Materials and Methods

A historical cohort consisting of all patients diagnosed with invasive breast cancer who received their entire treatment at the INC-C and were referred to genetic counseling as standard of care, between 1 April 2018 and 30 June 2021, were included in this study. A total of 912 newly diagnosed breast cancer patients were admitted to the Breast Unit in this period. Of these incident cases, 140 were referred to the Genetic Clinic and underwent germline testing based on the criteria established by the NCCN guidelines v2.2021 [18,19]. Germline testing was conducted on blood samples using a 105-gene hereditary cancer panel. The list of genes can be found in Appendix A. The testing was carried out in our laboratories using a MiSeq instrument (Illumina Inc., San Diego, CA, USA). A standardized and validated next-generation sequencing (NGS) method was implemented, with specific details and protocols referenced in prior publications from our group [20,21]. Out of the 105 cases with a PV identified in any gene, 23 were excluded from the study as they did not receive complete treatment at the INC-C. Consequently, the analysis of Disease-Free Survival (DFS) and Overall Survival (OS) outcomes was performed on a final cohort of 82 carriers with breast cancer who received comprehensive management and follow-up care at the INC-C Breast Unit. Ethical approval for this study was obtained from the Ethics Committee of the INC-C, ensuring compliance with ethical standards and patient privacy.

The sociodemographic and clinicopathological characteristics, as well as the genetic information, were retrieved from multiple sources, including the Breast Unit database, the electronic medical records system, SAP, and the Registry of the Institutional Hereditary Cancer Program. One author collected the data, which were then entered into a Research Electronic Data Capture (REDCap) platform. An assigned monitor from the Institutional Research Division implemented a data quality assessment. The variables analyzed here included age, clinical and pathological characteristics of the disease, reasons for referral to genetics based on current NCCN criteria, genetic testing results, family history of cancer in first- and second-degree relatives, details of medical and surgical treatments, implementation of risk-reducing measures, recurrence, and mortality attributed to breast cancer.

Descriptive statistics were obtained for categorical and nominal variables using measures of absolute and relative frequency. We evaluated DFS and OS as outcomes. We estimated DFS and OS using the Kaplan–Meier method. DFS was defined as the time from the initial diagnosis made during a multidisciplinary consultation, where both breast surgeons and oncologists are involved, to the identification of recurrence. OS was defined as the time from the initial diagnosis established during the multidisciplinary consultation to death from any cause. This endpoint was censored at loss to follow-up or the date of the last contact with the institution. The frequency of these outcomes was estimated using incidence rates, calculated as the number of events per hundred patient-years. Confidence intervals (CI) with a 95% level of certainty were reported for these rates. Statistical analyses were conducted using the software R-Project Statistical Computing Software version 4.2.1 (R Foundation for Statistical Computing, Vienna, Austria).

## 3. Results

### 3.1. Patient Characteristics

Genetic counseling referrals from the Breast Unit accounted for 15.4% of all incident breast cancer cases (140/912), and the prevalence of PVs was 11.5% (105/912). As mentioned before, the cohort analyzed here consisted of 82 carriers affected by breast cancer who were managed and followed at the INC-C Breast Unit. In this cohort, the median age at diagnosis was 46.0 years (interquartile range = 14.7), 62.2% (*n* = 51) were premenopausal, 51.2% (*n* = 42) belonged to the contributory healthcare scheme, 34.1% (*n* = 28) had some degree of obesity, and 90.2% (*n* = 74) had an ECOG performance status of 0 to 1 at the time of diagnosis.

The most common reason for referral to genetics was age at diagnosis of 45 years or younger (48.7%; *n* = 40), followed by two or more cases of breast cancer in first- or second-degree relatives (46.3%; *n* = 38), TNBC subtype diagnosed at age 60 or earlier (31.7%; *n* = 26), synchronous or metachronous cancers (13.4%; *n* = 11), family history of ovarian cancer (4.8%; *n* = 4), and family history of male breast cancer (2.4%; *n* = 2) (Table 1). It is important to note that many patients met more than one referral criterion. Regarding the clinicopathological characteristics, 62.2% (*n* = 51) of patients had locally advanced breast cancer, with a considerable proportion having a histological grade of 3 (47.6%, *n* = 39) and non-special histology, mainly ductal (90.2%, *n* = 74). TNBC was the most frequent subtype, with 35.4% (*n* = 29), while 13.4% (*n* = 11) of cases exhibited overexpression of HER2 receptor (Table 1).

Most patients had neoadjuvant chemotherapy as the initial treatment (71.9%; *n* = 59), predominantly based on anthracyclines and taxanes (48.8%; *n* = 40). Also, 92.7% (*n* = 76) of the patients underwent surgical treatment, with modified radical mastectomy being the most common approach (46.3%; *n* = 38), followed by quadrantectomy in combination with other surgical procedures (34.1%; *n* = 28) (Table 2).

### 3.2. Prevalence of Germline PVs in Cancer-Predisposing Genes

In the present study, breast cancer patients were categorized into two groups based on genetic testing results. The first group, referred to as “associated gene carriers”, consists of patients carrying PVs in genes known to confer a 17% or higher absolute risk of breast cancer, as per NCCN guidelines [5]. The second group, termed “non-associated gene carriers”, comprises cases with PVs in other genes not associated with breast cancer susceptibility and who are wild-type for associated genes.

Most of the 82 breast cancer patients were classified into the first group of “associated gene carriers”, and they were diagnosed with a hereditary breast cancer syndrome (74.4%; *n* = 61), referred to as inherited breast cancer cases. The gene *BRCA2* was the most frequently mutated (36.6%; *n* = 30), followed by the *BRCA1* gene (17.1%; *n* = 14), accounting for 53.7% of all PVs detected in this cohort and 72.1% (44/61) of the hereditary cases (Figure 1). Other homologous recombination DNA damage repair (HR-DDR) genes associated with breast cancer risk accounted for 15.9% (13/82) of the carriers, including *ATM* (*n* = 2), *BARD1* (*n* = 4), *CHEK2* (*n* = 1), *PALB2* (*n* = 2), *RAD51C* (*n* = 1), and *RAD51D* (*n* = 3). Four additional carriers corresponded to rare high-risk breast cancer syndromes involving neurofibromatosis type 1 (*NF1*; *n* = 1; 1.2%), Cowden syndrome (*PTEN*; *n* = 1; 1.2%), and Li–Fraumeni syndrome (*TP53*; *n* = 2; 2.4%).

The remaining patients (21/82) were combined into the second group of “non-associated gene carriers”, none of whom were diagnosed with a hereditary breast cancer syndrome and are referred to as other breast cancer cases. Among these cases, eight were incidental findings, as the mutated gene is associated with cancer risks different from breast cancer, such as Lynch syndrome (*n* = 5 *PMS2*), Birt–Hogg–Dube syndrome (*n* = 1 *FLCN*), susceptibility to malignant cutaneous melanoma (*n* = 1 *MITF*), and hereditary paraganglioma and pheochromocytoma (*n* = 1 *SDHB*). The remaining other breast cancer cases (13/21) were carriers of a heterozygous PV in genes associated with autosomal recessive inheritance syndromes (i.e., *MUTYH*, *FANCA*, *NTHL1*, *XPC*, *REQL4*, *BLM*, *NBN*, *BUB1B*, and *ERCC4*) and did not receive a diagnosis of a hereditary cancer syndrome. The most common type of PV was frameshift (50%; *n* = 41).

Regarding the distribution of germline PVs on genes associated with breast cancer risk according to the molecular subtypes determined by St. Gallen 2013 surrogates (Figure 1), we found that 78.6% of *BRCA1* carriers developed TNBC, whilst 53.3% of *BRCA2* carriers had the luminal B HER2-negative subtype.

A total of 15 of 49 inherited breast cancer cases in which a discussion to opt for risk-reduction mastectomy (RRM) was recommended given the mutated gene (i.e., *BRCA1*, *BRCA2*, *TP53*, *PALB2*, and *PTEN*) made the decision to undergo this procedure (30.6%). Among inherited cases with PVs on genes in which RRM is only recommended based on family history (i.e., *ATM*, *BARD1*, *CHEK2*, *RAD51C*, *RAD51D*, and *NF1*), one opted for this intervention (8.3%; 1/12). Regarding risk-reducing salpingo oophorectomy (RRSO), 28 of 50 inherited cases with PVs in genes that should prompt the recommendation for this procedure (i.e., *BRCA1*, *BRCA2*, *RAD51C*, *RAD51D*, and *PALB2*) accepted this intervention (56%), and an additional case with a *PTEN* PV also opted for this intervention.

### 3.3. Survival Outcomes and Disease Recurrence in Breast Cancer Patients Carrying PVs in “Associated”and “Non-Associated” Breast Cancer Genes

Most of the 28 patients who underwent quadrantectomy in combination with other surgical procedures corresponded to inherited breast cancer cases with PVs in associated genes (71.4%; 20/28), mainly *BRCA2* (9/20; 45%) and *BRCA1* (7/20; 35%). None of these patients experienced local or locally advanced recurrences; however, three cases did exhibit distant recurrence. Conversely, the rest of the patients managed with quadrantectomy (28.6%; 8/28) belong to the group of other breast cancer cases with PVs in non-associated genes, with only one experiencing distant recurrence. Among the 16 patients who underwent RRM, only 2 experienced recurrences. One patient had regional and distant recurrence in the liver and CNS, while the other patient had distant recurrence in the bones. Both cases were associated with a *BRCA2* PV.

Overall, 20 patients (24.4%) experienced disease recurrence, predominantly at distant sites (90.0%; *n* = 18), of which 15 were locally advanced. The most common sites of metastatic involvement were the central nervous system (CNS) (*n* = 9), followed by the lungs (*n* = 8) and bones (*n* = 6). Only one patient experienced local recurrence, with clinical stage IIB, HER2-enriched subtype and a PV in the *NF1* gene. This patient received prolonged neoadjuvant therapy and underwent modified radical mastectomy. Subsequently, the patient experienced distant recurrence (lungs and CNS), and, at the last contact, their status was known to be alive with the disease. Patients with disease progression (*n* = 20) were primarily carriers of PVs in the *BRCA2* gene (*n* = 7; 35.0%), of which three had luminal B HER2-negative tumors, while another 25% (*n* = 5) of these were carriers of PVs in the *BRCA1* gene, all of whom had triple-negative molecular subtypes.

The median follow-up time for the entire cohort was calculated to be 38.1 months. Specifically, patients categorized as “associated gene carriers” had a median follow-up of 36.7 months, while for those in the “non-associated gene carrier” group, this duration was 44.1 months. Patients within the first group (i.e., inherited breast cancer cases) exhibited a 5-year DFS rate of 66.5% (95% CI: 52.9–86.0), while for those within the second group (i.e., other breast cancer cases), the 5-year DFS was 88.2% (95% CI: 74.2–100) (Figure 2B). Moreover, the 5-year DFS for carriers of PVs in the *BRCA2* gene was 37.6% (95% CI: 14.1–100), for carriers of PVs in *BRCA1* it was 76.9% (95% CI: 57.1–100), and for carriers of PVs in other HR-DDR genes, it was 80.2% (95% CI: 65.4–100) (Figure 2D).

At the time of study closure, the majority of patients in the cohort were alive with no evidence of disease (*n* = 56; 68.3%). A total of 12 patients (14.6%) remained alive with active disease, while 10 patients (12.2%) had succumbed to breast cancer, 6 of whom presented advanced stages (IIIB, *n* = 4; IIIC, *n* = 1; IV, *n* = 1), and most carried PVs in the *BRCA1* (*n* = 4; 40.0%) and *BRCA2* (*n* = 3; 30.0%) genes. Two patients (2.4%) died from other causes. The mortality rate was 3.06 deaths per 100 patients/year (95% CI: 1.58–5.35). The 5-year OS for inherited breast cancer cases was 79.2% (95% CI: 66.2–94.7), and for the other breast cancer cases group, it was 89.4% (95% CI: 76.5–100) (Figure 2A). Among patients with a PV in the *BRCA2* gene, the 5-year OS was 68.8% (95% CI: 45.8–100), while for those with PVs in *BRCA1*, it was 73.1% (95% CI: 50.8–100). Patients with PVs in other HR-DDR genes had a 100% OS rate at 5 years (Figure 2C).

## 4. Discussion

Hereditary breast cancer represents a significant proportion of breast cancer cases worldwide, with varying prevalence rates reported across different populations. In this study, we investigated the characteristics and outcomes of patients with hereditary breast cancer in a clinic-based sample from Colombia. Our findings provide insights into the prevalence of germline PVs, clinicopathological features, treatment modalities, and survival outcomes in this high-risk population.

Few studies on Colombian breast/ovarian cancer cases have reported a yield of PVs of 11% to 22% in selected cases based on international criteria and using a multigene hereditary cancer panel, including 25 to 143 genes [15,16,17]. In our study, only 140 out of 912 newly diagnosed breast cancer cases seen at the Breast Unit of the INC-C between 2018 and 2021 were referred to genetic counseling, and 11.5% (105/912) were identified as carriers of a germline PV using a 105 multigene panel. Our findings are consistent with other studies reporting a prevalence of PVs between 5% and 15% among breast cancer cases [22,23,24,25,26].

The referral patterns for genetic counseling in our study were mainly based on age at diagnosis, family history, and molecular subtypes associated with hereditary breast cancer. These criteria are consistent with established guidelines and recommendations for genetic testing in patients with breast cancer. Nevertheless, recent studies have highlighted the limitations of current genetic testing guidelines in identifying germline variants in patients with solid tumor malignancies [27,28,29]. These studies suggest that many individuals with actionable germline variants are missed by current guidelines. This underscores the importance of expanding panel testing and incorporating tumor genetic profiling into clinical practice [27,28,29,30,31,32]. In the context of limited economic resources in Colombia, our work can serve as a starting point to raise awareness among healthcare providers and decision makers. Subsequent cost-effective studies may provide an opportunity to reevaluate the feasibility of implementing these approaches, ultimately improving the diagnosis and management of hereditary breast cancer in Colombia.

In this study, we observed that *BRCA2* is the most frequently mutated gene, followed by *BRCA1.* These findings are consistent with global data indicating that PVs in *BRCA* genes account for a significant proportion of hereditary breast cancer cases [3,16,33,34] and that a higher proportion of PVs in *BRCA2* are reported among these cases from Colombia [15], Puerto Rico [35], and Cuba [36], whereas *BRCA1* PVs are more frequent in Mexico [37,38], Argentina [39], Brazil [40], and Peru [41]. Other HR-DDR genes, such as *ATM*, *BARD1*, *CHEK2*, *PALB2*, *RAD51C*, and *RAD51D*, were also identified in our cohort, albeit at a lower frequency. Consistently, others have also reported an important contribution of PVs in HR-DDR genes in hereditary breast cancer cases in different Latin American and Hispanic populations [16,42].

Clinically, our cohort demonstrated aggressive tumor characteristics, with a significant proportion of locally advanced tumors and high-grade histology. The TNBC subtype was the most prevalent in our cohort of carriers of PVs, followed by luminal B HER2-negative. Notably, we observed that most cases with PVs in the *BRCA1* gene developed breast cancer of the triple-negative subtype, with grade III differentiation, and at a young age. Conversely, PVs in the *BRCA2* gene were predominantly found in luminal breast cancer cases. This is consistent with previous studies reporting the same phenotype–genotype correlations [5,19,23].

The decision to undergo RRM and RRSO is a complex consideration for individuals with a hereditary breast cancer syndrome unaffected or affected by breast cancer. Complications, such as bleeding, infection, flap necrosis, seroma formation, and pain, can arise in a significant proportion of cases (ranging from 30 to 64%). Hence, it is imperative that this choice is reached through shared decision making between the carrier and their healthcare provider and considering various factors, such as the mutated gene, the personal and/or family history of cancer, the risk perception, psychosocial factors, age, overall health, reproductive plans, and personal values [43,44].

Bilateral RRM confers a substantial reduction in breast cancer risk for unaffected individuals with PVs in *BRCA1–2* genes, estimated at 90% and up to 95% when combined with RRSO [45,46,47,48]. Regarding mortality risk, a recent population-based cohort study reported that it was lower among unaffected individuals with PVs in the *BRCA1* gene who underwent bilateral RRM compared to those under surveillance, though this effect was not observed in unaffected individuals with PVs in the *BRCA2* gene [47,48]. The annualized cumulative incidence rate of contralateral breast cancer among unilateral breast cancer cases with germline PVs in *BRCA1* and *BRCA2* genes is 2.3% and 1.7% per year, respectively [49,50]. Due to this increased risk, some may be considered for contralateral RRM, although the oncological benefits are not clear given that their prognosis is primarily dictated by their primary breast cancer [48,51,52]. This evidence underscores the need for more individualized counseling based on the *BRCA* variant type and the difficulties inherent in choosing between bilateral RRM and breast cancer surveillance among unaffected and affected carriers [47,48].

Among the 61 inherited breast cancer cases in this cohort, the overall rate of any risk-reducing surgery intake in these cases was 55.7% (34/61); specifically, 26.2% (16/61) had an RRM, 47.5% (29/61) had an RRSO, and 11 cases (18.1%) had both procedures. The uptake of RRM and RRSO varied depending on the mutated gene, with higher acceptance rates observed for carriers of PVs in genes with the strongest recommendations, such as *BRCA* genes. On the contrary, 39.3% did not undergo any of these procedures; as expected, the main reason for not having risk-reducing surgery was being a carrier of PVs in genes for which there is insufficient evidence regarding the impact on oncological outcomes. Another reason for not undergoing risk-reducing interventions was the presence of locally advanced or metastatic disease. This rationale is understandable, as most trials assessing the benefit of these procedures in patients with *BRCA1–2* PVs excluded women with advanced breast cancer. Recommendations for such cases, despite being treatable, are not fully supported, as they remain incurable, with a 5-year survival of 27.4%. Furthermore, the increased probability of death is primarily related to disease progression rather than the development of a second primary cancer [53,54].

Consistent with the high proportion of locally advanced breast cancer in our cohort, neoadjuvant chemotherapy was the primary treatment modality in our cohort, and surgical management primarily involved modified radical mastectomy. The utilization of anthracycline and taxane-based regimens reflects standard treatment protocols in this setting [55]. Currently, there are targeted therapies available for patients with germline PVs in *BRCA* genes. The OlympiaA study demonstrated the clinical benefit of olaparib for one year in the adjuvant setting, showing superior DFS compared to placebo (HR 0.58; 99.5% CI 0.41–0.82, *p* < 0.001) [56]. The impact of this novel therapy could not be evaluated in this cohort given that during the study period, the therapy had not yet received authorization for breast cancer treatment by the respective Colombian regulatory entity (i.e., The National Institute of Drug and Food Surveillance, INVIMA).

Breast-conserving surgery in carriers of PVs in the *BRCA1–2* genes has been associated with a greater risk of local recurrence compared to mastectomy, although it did not significantly impact patient survival in terms of DFS and OS [57]. Studies support that combining RRSO with breast-conserving surgery can reduce the risk of distant metastases and enhance survival [58], offering a viable option for young *BRCA*-carriers with small breast cancers to avoid up-front mastectomy. RRSO could be considered when the epidemiological risk of ovarian cancer increases and reproductive desires are fulfilled [58]. In our study, none of the breast cancer patients with germline PVs and managed with quadrantectomy (*n* = 28) experienced local or locally advanced recurrences, although four had distant recurrences.

Survival outcomes and disease recurrence are critical factors in evaluating the long-term impact of hereditary breast cancer. Previous studies with substantial heterogeneity in the selection and classification of the carriers and non-carriers groups have led to conflicting results regarding a worse survival for cases carrying PVs in *BRCA1* and *BRCA2* genes; nevertheless, a tendency towards a survival disadvantage for all outcomes was observed in a systematic review and meta-analysis study of individuals with PVs in *BRCA* genes [59]. Our study echoes this trend, as evidenced by our findings on DFS and OS rates among groups categorized by PVs in associated genes (i.e., *BRCA1*, *BRCA2,* and HR-DDR or inherited breast cancer cases) and those with PVs in non-associated genes (i.e., other breast cancer cases). By examining these specific categories, our study aimed to provide insight into the prognostic implications of different genetic profiles among breast cancer patients.

This study has limitations. Being an observational descriptive cohort study, the quality of information is contingent upon the precision and accuracy of data recording. The sample size is relatively small, and, given the descriptive nature of the design, this study does not establish a formal comparison between groups. The follow-up duration may not accurately capture long-term outcomes and survival rates. Finally, because DFS and OS were estimated without adjustment for the presence of confounding variables, such as age, molecular subtype, and chemotherapy treatment, the observed results may be susceptible to residual confusion.

This study also has several strengths. It was conducted at the INC-C, a highly specialized institution and the country’s largest reference cancer center known for its expertise in cancer care. Within this institutional environment, cancer patients receive comprehensive, multidisciplinary care following institutional guidelines that align with international standards, thereby providing a robust foundation for our research. Employing a validated NGS sequencing analysis, we systematically identified genetic variants across our cohort using a standardized multigene panel of 105 genes. This uniform approach ensured consistent genetic testing for all patients at our genetic laboratories, facilitating accurate variant interpretation through a well-established protocol. Additionally, our ongoing Institutional Hereditary Cancer Program played a pivotal role in supporting clinicians in interpreting genetic testing results, particularly within the context of patients’ personal and family history. This support is especially crucial when navigating the complexities of large multigene panels, which pose challenges in identifying variants of uncertain significance and unexpected genetic findings, including cases with monoallelic pathogenic variants associated with autosomal recessive conditions. Furthermore, our research sheds light on the most prevalent genes with PVs across various molecular breast cancer subtypes, enhancing our understanding of phenotype–genotype relationships within our study cohort. These strengths not only offer practical guidance for healthcare professionals but also deepen our comprehension of hereditary breast cancer in Colombians, underscoring the clinical significance of our findings.

## 5. Conclusions

Our study highlights the prevalence of germline PVs in hereditary breast cancer and their impact on clinicopathological features, treatment decisions, and survival outcomes. Genetic counseling and testing play a crucial role in identifying individuals at high risk and guiding personalized management decisions. Further research is warranted to explore novel therapeutic approaches and improve survival outcomes for patients with hereditary breast cancer in the Colombian context.

## Figures and Tables

**Figure 1 cancers-16-02020-f001:**
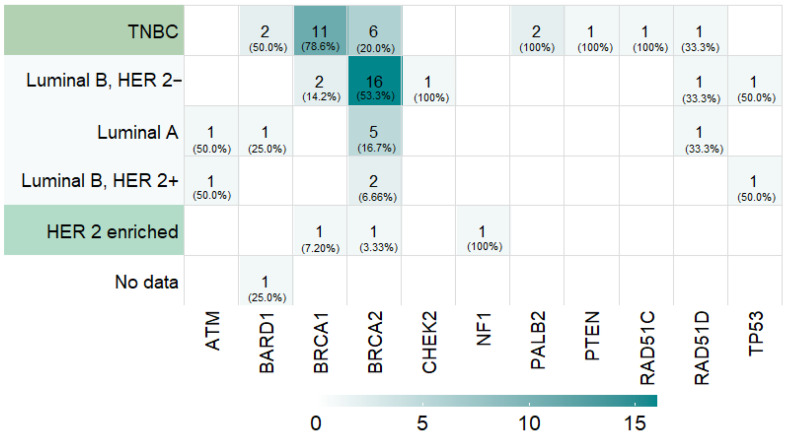
PVs identified in genes associated with hereditary breast cancer syndromes and stratified by the molecular subtype according to the St. Gallen 2013 surrogate classification.

**Figure 2 cancers-16-02020-f002:**
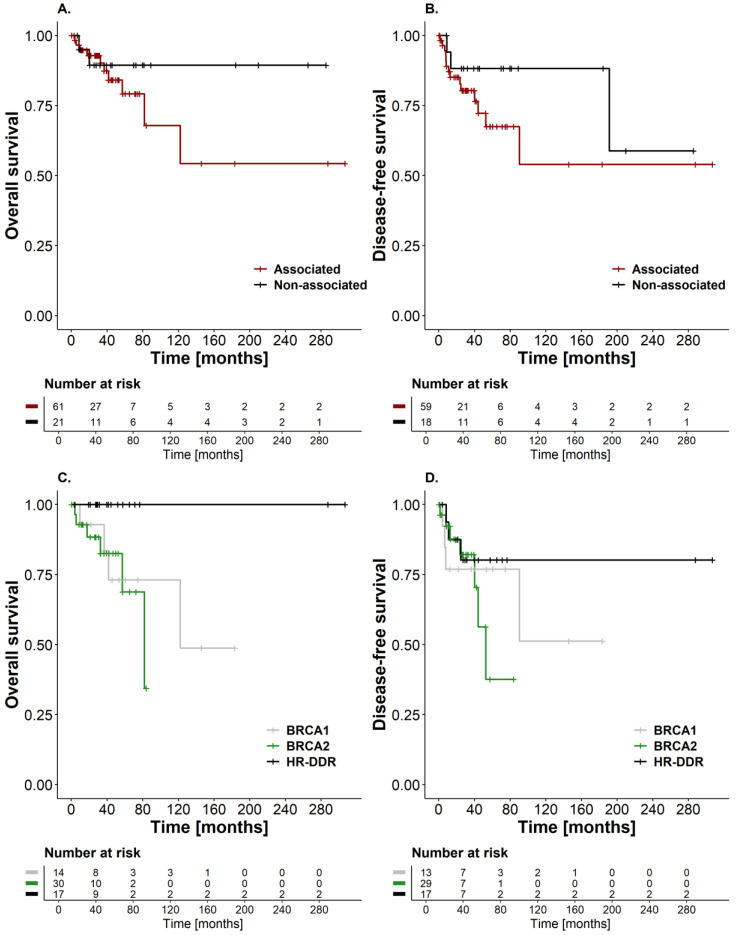
Kaplan–Meier curves for (**A**) Overall Survival and (**B**) Disease-Free Survival in “associated gene carriers” and “non-associated gene carriers”. (**C**) Overall Survival and (**D**) Disease-Free Survival differences among inherited breast cancer cases in relation to the mutated gene (*BRCA1*, *BRCA2*, and HR-DDR).

**Table 1 cancers-16-02020-t001:** Clinical and pathological characteristics of 82 carriers affected by breast cancer.

Characteristics	Count (*n*)	Percentage (%)
Age (years)		
≤40	26	31.7
41–50	26	31.7
51–60	23	28.0
>60	7	8.5
Indications for referral to genetics		
Age ≤ 45 years	40	48.7
TNBC ≤ 60 years	26	31.7
≥2 breast cancer cases in FDR or SDR	38	46.3
Metachronous and/or synchronous cancers	11	13.4
Family history of ovarian cancer	4	4.8
Male family member with breast cancer	2	2.4
Family history of prostate cancer *	2	2.4
Clinical stage		
In situ	2	2.4
IA	8	9.8
IB	1	1.2
IIA	14	17.1
IIB	13	15.9
IIIA	7	8.5
IIIB	27	32.9
IIIC	4	4.9
IV	5	6.1
Histological type		
Ductal (NST)	74	90.2
Lobular	2	2.4
Medullary	2	2.4
Metaplastic	1	1.2
Other (papillary)	1	1.2
No data	2	2.4
Histological grade		
I	3	3.7
II	33	40.2
III	39	47.6
No data	7	8.5
St. Gallen 2013 surrogate classification		
Luminal A	13	15.8
Luminal B, HER 2−	25	30.5
Luminal B, HER 2+	6	7.3
TNBC	29	35.4
HER 2 enriched	5	6.1
No data	4	4.9

FDR: first-degree relative; SDR: second-degree relative; NST: No Special Type; TNBC: triple-negative breast cancer. * Based on NCCN guidelines criteria.

**Table 2 cancers-16-02020-t002:** Treatment modalities administered to the patients in the study cohort.

Treatment	Count (*n*)	Percentage (%)
Neoadjuvant chemotherapy	59	71.9
AC-T	40	48.8
AC-TCb	6	7.3
AC-TH	5	6.1
AC alone	4	4.9
Other	4	4.9
Surgery of the primary tumor	76	92.6
Quadrantectomy + sentinel lymph node *	10	12.2
Quadrantectomy + axillary lymph node dissection *	18	22
Modified radical mastectomy	38	46.3
Simple mastectomy + sentinel lymph node	8	9.8
Other	2	2.4
Adjuvant chemotherapy	25	30.4
AC-T	4	4.9
AC-TH	2	2.4
AC alone	4	4.9
Adjuvant capecitabine	8	9.8
Adjuvant trastuzumab	5	6.1
Other	2	2.4
Adjuvant radiotherapy	58	70.7
3DCRT	41	50
IMRT	17	20.7
Adjuvant hormone therapy	39	47.5
Tamoxifen	26	31.7
Tamoxifen + aromatase inhibitor	9	11
Aromatase inhibitor	2	2.4
GNRH analogues + tamoxifen	2	2.4

AC-T: anthracyclines and taxanes; AC-TCb: anthracyclines, taxanes, and Carboplatins; AC-TH: anthracyclines, taxanes, and trastuzumab; AC: anthracyclines; 3DCRT: Three-Dimensional Conformal Radiotherapy; IMRT: Intensity Modulated Radiation Therapy; GNRH: Gonadotropin-Releasing Hormone. * Including oncoplastic breast surgery.

## Data Availability

The data that support the findings of this study are available from the corresponding author upon reasonable request.

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
