# Peer review of "Management and Clinical Outcomes of Breast Cancer in Women Diagnosed with Hereditary Cancer Syndromes in a Clinic-Based Sample from Colombia"

_cancers, 2024, doi:10.3390/cancers16112020_

Round 1

Reviewer 1 Report

Comments and Suggestions for Authors

In this study, the authors reported their mono-institutional experience of follow-up at the Breast Unit of Instituto Nacional de Cancerologia in Colombia of breast cancer women who underwent germline genetic testing by a 105-gene hereditary cancer panel and who resulted in carriers of a mutation.

The authors stated that the study aimed to investigate prognosis and survival differences in these patients.

The cohort analyzed by this study consisted of 82 patients.

Out of 82 patients, 61 were carriers of a mutation in a hereditary breast cancer gene.

The remaining patients (n=21) were carriers of a mutation in a gene not strictly associated with breast cancer.

The authors aimed to analyze the differences between these two groups.

However, the sample size appears to be too limited for the intention of investigating prognosis and survival. 

In addition, it would have been interesting to observe the differences between carriers of mutations in breast cancer predisposing genes compared to patients found to be wild-type much more than compared to patients carrying mutations in other genes whose role in breast cancer occurrence is not yet fully elucidated (PMS2, FLCN, etc.).

The term mutation is too general. The authors introduced the P/LP acronym in the introduction. Using these acronyms throughout the text, replacing the term mutation, is recommended.

The section "Material and Methods" could be more detailed. For example, a list of the genes included in the hereditary cancer panel (also as supplementary material) could be helpful. In the methods, the authors could improve the definition of OS and provide more details about the duration and the modality of follow-up. Indeed, it could be interesting for the readers to know how this Breast Unit followed up breast cancer patients with germline P/LP, both those with germline P/LP in breast cancer predisposition genes and other hereditary cancer predisposition genes.  

Some conclusions seem a bit hasty considering the study's limitations (e.g., BRCA2 patients who would have a worse prognosis than BRCA1 patients). Finally, the authors should point out what the originality of their study is, given the limitations that can not allow for novel evidence on prognosis. For example, the authors could consider focusing on the results obtained by germline genetic testing and the challenges in managing multigene panel test results (e.g., secondary findings or heterozygous P/LP in genes associated with autosomal recessive conditions).

Author Response

Comments and Suggestions for Authors

  • In this study, the authors reported their mono-institutional experience of follow-up at the Breast Unit of Instituto Nacional de Cancerologia in Colombia of breast cancer women who underwent germline genetic testing by a 105-gene hereditary cancer panel and who resulted in carriers of a mutation.

The authors stated that the study aimed to investigate prognosis and survival differences in these patients.

The cohort analyzed by this study consisted of 82 patients.

Out of 82 patients, 61 were carriers of a mutation in a hereditary breast cancer gene.

The remaining patients (n=21) were carriers of a mutation in a gene not strictly associated with breast cancer.

The authors aimed to analyze the differences between these two groups.

However, the sample size appears to be too limited for the intention of investigating prognosis and survival. 

Response:

We appreciate the valuable input provided by the reviewer. Nevertheless, it is important to note that the proposed design corresponds to a descriptive study of a historical cohort, aimed primarily at describing disease-free survival (DFS) and overall survival (OS) among a cohort of breast cancer patients carrying germline pathogenic/likely pathogenic variants (PVs). Given its non-analytical nature (no hypothesis testing is intended), the calculation of a sample size a priori is not warranted (as is typically done in other methodological approaches).

This statement is outlined and edited in the Methods section (page 3, paragraph 3): “Descriptive statistics were obtained for categorical and nominal variables using measures of absolute and relative frequency. We evaluated DFS and OS as outcomes. We estimated DFS and OS using the Kaplan-Meier method. DFS was defined as the time from the initial diagnosis made during a multidisciplinary consultation, where both breast surgeons and oncologists are involved, to the identification of recurrence. OS was defined as the time from the initial diagnosis established during the multidisciplinary consultation, until death from any cause. This endpoint was censored at loss to follow-up or the date of the last contact with the institution. The frequency of these outcomes was estimated using incidence rates, calculated as the number of events per hundred patient-years.”

  • In addition, it would have been interesting to observe the differences between carriers of mutations in breast cancer predisposing genes compared to patients found to be wild-type much more than compared to patients carrying mutations in other genes whose role in breast cancer occurrence is not yet fully elucidated (PMS2, FLCN, etc.).

Response:

Thank you for your thoughtful suggestion. In this descriptive study of a historical cohort, aimed primarily at delineating DFS and OS among a cohort of breast cancer patients carrying germline PVs, we focused on describing outcomes in cases diagnosed with a hereditary breast cancer syndrome (i.e., meaning that are carriers of PVs in genes with a stablished strong association with breast cancer risk).

Breast cancer cases that did not received a diagnosis of a hereditary breast cancer syndrome (i.e., those with PVs in genes not known to confer increased risk for breast cancer) were categorized as a separate group with sporadic breast cancer. This group have wild-type BRCA1, BRCA2 and HR-DDR genes, since all cases were sequenced with an NGS multigene panel (for 105 genes), and no PVs were found in these genes.

Importantly, we categorize our cohort of patients in the groups mentioned, following NCCN guidelines (v2.2024 cited in the manuscript and concordant with the latest v3. 2024). By examining these specific categories, our study aimed to shed light on the prognostic implications for different genetic profiles among breast cancer patients.

While comparing carriers of PVs in breast cancer predisposing genes to individuals with negative genetic testing results could offer interesting insights, it falls outside the scope of our objectives planned for this descriptive study. We will consider exploring these comparisons in future investigations where appropriate.

  • The term mutation is too general. The authors introduced the P/LP acronym in the introduction. Using these acronyms throughout the text, replacing the term mutation, is recommended.

Response:

Thank you for your feedback. We appreciate your suggestion to use the P/LP (Pathogenic/Likely Pathogenic) acronym consistently throughout the manuscript, and to avoid the term “mutation”.

Upon careful consideration, we have made the necessary revisions throughout the manuscript to consistently utilize the equivalent abbreviation, PVs (for pathogenic/likely pathogenic variants).

  • The section “Material and Methods” could be more detailed. For example, a list of the genes included in the hereditary cancer panel (also as supplementary material) could be helpful.

Response:

We appreciate this suggestion. In accordance, we created a separate file for this supplement and added the following sentence to the Methods section (page 3, paragraph 1): The list of genes can be found in the Supplementary Table 1.

  • In the methods, the authors could improve the definition of OS and provide more details about the duration and the modality of follow-up.

Response:

Thank you for your valuable feedback. We appreciate the suggestion to enhance the definition of OS. We defined OS as the elapsed time from the date of diagnosis during multidisciplinary consultation (which involves participation from breasts surgeons and oncologists) to death from any cause. This endpoint was censored at loss to follow-up or the date of the last contact with the institution.

We incorporated these clarifications in the revised Methods section (page 3, paragraph 3): “Descriptive statistics were obtained for categorical and nominal variables using measures of absolute and relative frequency. We evaluated DFS and OS as outcomes. We estimated DFS and OS using the Kaplan-Meier method. DFS was defined as the time from the initial diagnosis made during a multidisciplinary consultation, where both breast surgeons and oncologists are involved, to the identification of recurrence. OS was defined as the time from the initial diagnosis established during the multidisciplinary consultation, until death from any cause. This endpoint was censored at loss to follow-up or the date of the last contact with the institution. The frequency of these outcomes was estimated using incidence rates, calculated as the number of events per hundred patient-years.”

  • Indeed, it could be interesting for the readers to know how this Breast Unit followed up breast cancer patients with germline P/LP, both those with germline P/LP in breast cancer predisposition genes and other hereditary cancer predisposition genes.

Response:

Thank you for raising this point. It is important to highlight that this study was conducted at the Instituto Nacional de Cancerología from Colombia (INC-C), the country’s largest cancer center known for its expertise in cancer care. At our Breast Unit, patients with hereditary cancer syndromes, identified as carriers of clinically actionable PVs with clear screening recommendations, undergo follow-up in collaboration with relevant specialized services within the same institution. This follow-up is customized based on their specific gene and guided by NCCN guidelines with respect to other cancer risks, besides contralateral breast cancer.

It's important to note that our study does not aim to assess the impact of different strategies for follow-up on survival outcomes, but rather focuses on describing the characteristics and outcomes of patients with germline PVs in established breast cancer genes “inherited cases” and those carrying PVs other in genes not associated with breast cancer risk (according to NCCN guidelines).

  • Some conclusions seem a bit hasty considering the study’s limitations (e.g., BRCA2 patients who would have a worse prognosis than BRCA1 patients).

Response:

We appreciate the reviewer’s thoughtful evaluation of our study findings. We edited our conclusions in accordance with our descriptive study design and observed outcomes within our cohort.

We added a new clarification at the end of the Discussion section (page 10, paragraph 4): “Previous studies with substantial heterogeneity in the selection and classification of the carriers and non-carriers groups, have led to conflicting results regarding a worse survival for cases carrying PVs in BRCA1 and BRCA2 genes, nevertheless, a tendency towards a survival disadvantage for all outcomes was observed in a systematic review and meta-analysis study of individuals with PVs in BRCA genes [57]. Our study echoes this trend, as evidenced by our findings on DFS and OS rates among groups categorized by PVs in associated genes (i.e., BRCA1, BRCA2 and HR-DDR or “inherited cases) and those with PVs in non-associated genes (i.e., “sporadic cases”). The inclusion of relevant cofounder variants, such as age, molecular subtype, and chemotherapy treatment, in further comparative models may shed light on potential disparities in tumor behavior and therapeutic response, particularly concerning BRCA-associated disease.”

  • Finally, the authors should point out what the originality of their study is, given the limitations that can not allow for novel evidence on prognosis. For example, the authors could consider focusing on the results obtained by germline genetic testing and the challenges in managing multigene panel test results (e.g., secondary findings or heterozygous P/LP in genes associated with autosomal recessive conditions).

Response:

Thank you for your insightful suggestion. We appreciate the opportunity to further clarify the originality of our study, particularly considering the identified limitations. Despite these constraints, our study possesses strengths and implications for clinical practice within our population.

We extend the description of Limitations and added a description of the Strengths of our study at the end of the Discussion section (page 10, paragraphs 5-6): This study has limitations. Being an observational descriptive cohort study, the quality of information is contingent upon the precision and accuracy of data recording. The sample size is relatively small and given the descriptive nature of the design this study does not establish a formal comparison between groups. The follow-up duration may not accurately capture long-term outcomes and survival rates. Finally, since DFS and OS were estimated without adjustment for the presence of confounding variables, the observed results may be susceptible to residual confusion.”

“This study has several strengths. It was conducted at the INC-C, a highly specialized institution and the country’s largest reference cancer center known for its expertise in cancer care. Within this institutional environment, cancer patients receive comprehensive, multidisciplinary care following institutional guidelines that align with international standards, thereby providing a robust foundation for our research. Employing a validated NGS sequencing analysis, we systematically identified genetic variants across our cohort using a standardized multigene panel of 105 genes. This uniform approach ensured consistent genetic testing for all patients at our genetic laboratories, facilitating accurate variant interpretation through a well-established protocol. Additionally, our ongoing Institutional Hereditary Cancer Program played a pivotal role in supporting clinicians in interpreting genetic testing results, particularly within the context of patients’ personal and family history. This support is especially crucial when navigating the complexities of large multigene panels, which pose challenges in identifying variants of uncertain significance and unexpected genetic findings, including cases with monoallelic pathogenic variants associated with autosomal recessive conditions. Furthermore, our research sheds light on the most prevalent genes with PVs across various molecular breast cancer subtypes, enhancing our understanding of phenotype-genotype relationships within our study cohort. These strengths not only offer practical guidance for healthcare professionals but also deepen our comprehension of hereditary breast cancer in Colombians, underscoring the clinical significance of our findings.”

In addition, we edited sections in the Simple Summary and the Abstract, to reflect the changes made in the manuscripts as a result of our acknowledgement of Limitations and Strengths of our study. There are at the bottom of each section:

Edits in the Simple Summary (page 1): “We described the distribution of germline PVs among the different breast cancer molecular subtypes in this cohort of carriers. We observed that most PVs were in known breast cancer susceptibility genes, and PVs in BRCA2 were the most frequent. Notably, we observed that patients with disease progression were predominantly carriers of PVs in BRCA2 gene.” 

Edits in the Abstract (page 1-2): “The 5-year Disease-Free Survival (DFS) for inherited breast cancer cases was 66.5%, and for sporadic cases, 88.2%. Particularly, for carriers of PVs in BRCA2 gene this was 37.6%. The 5-year Overall Survival (OS) rates ranged from 68.8% for those with PVs in BRCA2 to 100% for those with PVs in other HR-DDR genes. Further studies are crucial for understanding tumor behavior and therapy response differences among Colombian breast cancer patients with germline PVs.”

Reviewer 2 Report

Comments and Suggestions for Authors

‘Management and clinical outcomes of breast cancer women diagnosed with hereditary cancer syndromes in a clinic-based sample from Colombia’ is original article (manuscript) whose aim was to investigate prognosis and survival differences in breast cancer patients with germline mutations. Authors explored characteristics and outcomes of 82 breast cancer patients treated and monitored at the same clinic.

Manuscript is written in a well structured manner. All the cited references are relevant for the field and relatively new. Ethics statement is adequate. Results are presented in several tables and figures. Data presented is easy to understand. Conclusions are consistent with the evidence and arguments presented. They have showed the usefulness of genetic testing in identifying high risk individuals and in personalized care.

Authors have stated limitations of their study such as small sample size and retrospective design. Quality of English used is fine.

Therefore, I recommend this article to be accepted in present form.

Author Response

Comments and Suggestions for Authors

  • ‘Management and clinical outcomes of breast cancer women diagnosed with hereditary cancer syndromes in a clinic-based sample from Colombia’ is original article (manuscript) whose aim was to investigate prognosis and survival differences in breast cancer patients with germline mutations. Authors explored characteristics and outcomes of 82 breast cancer patients treated and monitored at the same clinic.

Manuscript is written in a well structured manner. All the cited references are relevant for the field and relatively new. Ethics statement is adequate. Results are presented in several tables and figures. Data presented is easy to understand. Conclusions are consistent with the evidence and arguments presented. They have showed the usefulness of genetic testing in identifying high risk individuals and in personalized care.

Response:

We are grateful for the reviewer’s positive evaluation of our manuscript’s structure, relevance of references, and clarity of data presentation. Thank you for your valuable feedback.

  • Authors have stated limitations of their study such as small sample size and retrospective design. Quality of English used is fine.

Response:

Thank you for your feedback. We appreciate your comments about the limitations section. Through this review process we believe that we can extend more this section, and we also added a strengths section.

We extend the description of Limitations and added a description of the Strengths of our study at the end of the Discussion section (page 10, paragraphs 5-6): This study has limitations. Being an observational descriptive cohort study, the quality of information is contingent upon the precision and accuracy of data recording. The sample size is relatively small and given the descriptive nature of the design this study does not establish a formal comparison between groups. The follow-up duration may not accurately capture long-term outcomes and survival rates. Finally, since DFS and OS were estimated without adjustment for the presence of confounding variables, the observed results may be susceptible to residual confusion.”

“This study has several strengths. It was conducted at the INC-C, a highly specialized institution and the country’s largest reference cancer center known for its expertise in cancer care. Within this institutional environment, cancer patients receive comprehensive, multidisciplinary care following institutional guidelines that align with international standards, thereby providing a robust foundation for our research. Employing a validated NGS sequencing analysis, we systematically identified genetic variants across our cohort using a standardized multigene panel of 105 genes. This uniform approach ensured consistent genetic testing for all patients at our genetic laboratories, facilitating accurate variant interpretation through a well-established protocol. Additionally, our ongoing Institutional Hereditary Cancer Program played a pivotal role in supporting clinicians in interpreting genetic testing results, particularly within the context of patients’ personal and family history. This support is especially crucial when navigating the complexities of large multigene panels, which pose challenges in identifying variants of uncertain significance and unexpected genetic findings, including cases with monoallelic pathogenic variants associated with autosomal recessive conditions. Furthermore, our research sheds light on the most prevalent genes with PVs across various molecular breast cancer subtypes, enhancing our understanding of phenotype-genotype relationships within our study cohort. These strengths not only offer practical guidance for healthcare professionals but also deepen our comprehension of hereditary breast cancer in Colombians, underscoring the clinical significance of our findings.”

  • Therefore, I recommend this article to be accepted in present form.

Response:

We find your positive assessment of our work encouraging, and we are deeply thankful.

Reviewer 3 Report

Comments and Suggestions for Authors

The study entitled "Management and clinical outcomes of breast cancer women diagnosed with hereditary cancer syndromes in a clinic-based sample from Colombia" describes the management and clinical outcomes of breast cancer (BC) patients with hereditary cancer syndromes in Colombia, focusing on those referred for genetic cancer risk assessment.

Out of 912 cases, 140 underwent genetic counseling, with an 11.5% prevalence of germline mutations identified via a 105-gene panel.

Key findings include a 5-year DFS of 66.5% for mutation carriers ("inherited cases") versus 88.2% for those with mutations in non-associated genes ("sporadic cases").

Congratulations to the authors of the study, I have few comments:

- "Breast Unit of the Instituto Nacional de Cancerología, Colombia (INC-C) between 2018 and 2021, as part of the implementation of an Institutional Hereditary Cancer Program." this sentence does not belong into the Introduction section and should be moved to the Materials and Methods;

- The main limitation of the study is the small sample size. Did you perform a power analysis? If no, why?

- To expand your Discussion section and improve the quality of your manuscript, please add this research PMID: 35534308, which investigates into the surgical management of BRCA-mutation carriers;

- In your conclusions you state "Genetic counseling and testing play a crucial role in identifying individuals at high risk and guiding personalized management decisions.". How do you guide personalized management decisions? Please elaborate.

Author Response

Comments and Suggestions for Authors

  • The study entitled "Management and clinical outcomes of breast cancer women diagnosed with hereditary cancer syndromes in a clinic-based sample from Colombia" describes the management and clinical outcomes of breast cancer (BC) patients with hereditary cancer syndromes in Colombia, focusing on those referred for genetic cancer risk assessment.

Out of 912 cases, 140 underwent genetic counseling, with an 11.5% prevalence of germline mutations identified via a 105-gene panel.

Key findings include a 5-year DFS of 66.5% for mutation carriers ("inherited cases") versus 88.2% for those with mutations in non-associated genes ("sporadic cases").

Congratulations to the authors of the study, I have few comments:

Response:

Thank you for your thoughtful feedback on our study. We're glad to hear that you found our research valuable. We appreciate your suggestions, and we are addressing them to improve the overall quality of our manuscript.

  • - "Breast Unit of the Instituto Nacional de Cancerología, Colombia (INC-C) between 2018 and 2021, as part of the implementation of an Institutional Hereditary Cancer Program." this sentence does not belong into the Introduction section and should be moved to the Materials and Methods;

Response:

Thank you for bringing this to our attention. We have made the necessary adjustment and reformulated this paragraph of the Introduction section (page 2, paragraph 4): In this study, we aimed to describe the clinicopathological features, prognosis, and survival rates of newly diagnosed breast cancer patients, managed and followed-up within the same institution, who were identified as carriers of germline PVs through a genetic cancer risk assessment program. Moreover, we provide information on the distribution of these genetic findings across different molecular subtypes, contributing to the characterization of the Colombian breast cancer patient population.”

  • - The main limitation of the study is the small sample size. Did you perform a power analysis? If no, why?

Response:

We appreciate the valuable input provided by the reviewer. Nevertheless, it is important to note that the proposed design corresponds to a descriptive study of a historical cohort, aimed primarily at describing disease-free survival (DFS) and overall survival (OS) among a cohort of breast cancer patients carrying germline pathogenic/likely pathogenic variants (PVs). Given its non-analytical nature (no hypothesis testing is intended), the calculation of a sample size a priori is not warranted (as is typically done in other methodological approaches).

This statement is outlined and edited in the Methods section (page 3, paragraph 3): “Descriptive statistics were obtained for categorical and nominal variables using measures of absolute and relative frequency. We evaluated DFS and OS as outcomes. We estimated DFS and OS using the Kaplan-Meier method. DFS was defined as the time from the initial diagnosis made during a multidisciplinary consultation, where both breast surgeons and oncologists are involved, to the identification of recurrence. OS was defined as the time from the initial diagnosis established during the multidisciplinary consultation, until death from any cause. This endpoint was censored at loss to follow-up or the date of the last contact with the institution. The frequency of these outcomes was estimated using incidence rates, calculated as the number of events per hundred patient-years.”

  • - To expand your Discussion section and improve the quality of your manuscript, please add this research PMID: 35534308, which investigates into the surgical management of BRCA-mutation carriers;

Response:

Thank you for your suggestion to consider incorporating the research PMID: 35534308 into our Discussion section. This study's focus on the surgical management of BRCA-mutation carriers aligns well with our research, enhancing the quality and relevance of our manuscript.

We included this reference and expanded our Discussion around this topic (page 10, paragraph 3): “Breast-conserving surgery in carriers of PVs in the BRCA1-2 genes has been associated with a greater risk of local recurrence compared to mastectomy, although it did not significantly impact patient survival in terms of DFS and OS OS [57]. Studies support that combining RRSO with breast-conserving surgery can re-duce the risk of distant metastases and enhance survival [58], offering a viable option for young BRCA-carriers with small breast cancers to avoid up-front mastectomy. RRSO could be considered when the epidemiological risk of ovarian cancer increases, and reproductive desires are fulfilled [58]. In our study, none of the breast cancer patients with germline PVs and managed with quadrantectomy (n = 28) experienced local or locally advanced recurrences, although four had distant recurrences.”

  • - In your conclusions you state "Genetic counseling and testing play a crucial role in identifying individuals at high risk and guiding personalized management decisions.". How do you guide personalized management decisions? Please elaborate.

Response:

Thank you for raising this important point. In our study, personalized management decisions are guided by a multidisciplinary approach within the institutional framework of the INC-C, Colombia’s largest reference cancer center. Through our comprehensive Hereditary Cancer Program, which involves collaboration among breast surgeons, oncologists, and experts in genetic cancer risk assessment, we ensure that each patient receives tailored, individualized care, based on the genetic test result. Importantly, as a highly specialized institution for cancer care, patients are referred to other specialized units as needed for further evaluation and follow-up/screening. For instance, patients with Lynch syndrome are closely managed by specialists in gastrointestinal and gynecology services at our institution.

We elaborate this in the new added strengths section at the end of the Discussion (page 10, paragraph 6): “This study has several strengths. It was conducted at the INC-C, a highly specialized institution and the country's largest reference cancer center known for its expertise in cancer care. Within this institutional environment, cancer patients receive comprehensive, multidisciplinary care following institutional guidelines that align with international standards, thereby providing a robust foundation for our research. Employing a validated NGS sequencing analysis, we systematically identified genetic variants across our cohort using a standardized multigene panel of 105 genes. This uniform approach ensured consistent genetic testing for all patients at our genetic laboratories, facilitating accurate variant interpretation through a well-established protocol. Additionally, our ongoing Institutional Hereditary Cancer Program played a pivotal role in supporting clinicians in interpreting genetic testing results, particularly within the context of patients' personal and family history. This support is especially crucial when navigating the complexities of large multigene panels, which pose challenges in identifying variants of uncertain significance and unexpected genetic findings, including cases with monoallelic pathogenic variants associated with autosomal recessive conditions. Furthermore, our research sheds light on the most prevalent genes with PVs across various molecular breast cancer subtypes, enhancing our understanding of phenotype-genotype relationships within our study cohort. These strengths not only offer practical guidance for healthcare professionals but also deepen our comprehension of hereditary breast cancer in Colombians, underscoring the clinical significance of our findings.”

Round 2

Reviewer 1 Report

Comments and Suggestions for Authors

Firstly, thank the authors for exhaustively responding to the comments and following the various suggestions.

The purpose of the study thus appears adequately illustrated, and the manuscript improved.

Unfortunately, it seems that the authors cannot improve it with a further comparison with patients who are wild-type and so more correctly definable as "sporadic breast cancers."

This analysis would have further paid tribute to the potential of this study.

In light of the authors' answers, further minor comments:

-The term "sporadic breast cancers" may be misleading. Something like "breast cancer cases carrying PVs in other cancer predisposition genes" could avoid misinterpretations. Or, more simply, they could avoid the term "sporadic breast cancers" in all the text.

- In the introduction (in the third paragraph), the authors point out that "Despite these reports, there is insufficient characterization regarding the impact of BRCA and non-BRCA germline PVs on the survival and outcomes among the population affected with hereditary breast cancer syndromes in the Columbian context." 

However, they stated that they categorized two different groups: breast cancer patients carrying germline PVs in genes with an established association with breast cancer risk (including patients with moderate penetrance non-BRCA genes, such as ATM, CHEK2, and others) and those carrying PVs in genes not known to confer increased risk for breast cancer (they defined them "sporadic breast cancer"). 

To improve consistency and clarity, the authors could add a sentence to the introduction (last paragraph) clarifying that they will not compare BRCA and non-BRCA cases (indeed, they do it between BRCA+other breast cancer predisposition genes vs. other genes).

-In the methods, the authors could provide the average follow-up duration for all patients, both for those with mutations in genes associated with breast cancer risk and for patients with sporadic breast cancer. 

-The authors responded exhaustively to the comment about the interest in observing the difference between breast cancer patients carrying germline PVs in genes with an established association with breast cancer risk and breast cancer patients with uninformative results. They could add a sentence about this issue to the discussion. 

Author Response

Response to reviewer comments

Manuscript ID cancers-2969702

Management and clinical outcomes of breast cancer women diagnosed with hereditary cancer syndromes in a clinic-based sample from Colombia.

Dear Editor and reviewers,

We sincerely thank you for the opportunity to revise our manuscript based on the insightful feedback provided during the initial review process. We have carefully considered all the comments and suggestions from Reviewer 1, for this second round of revisions, and have made the necessary adjustments accordingly. Attached below are detailed responses to all the reviewer’s comments. The latter are shown in black and our responses in red. Please let us know if you still have any questions or concerns about the manuscript. We are committed to addressing any remaining issues to ensure the quality and clarity of our work.

Second Round - Reviewer 1

Review Report Form

Open Review

Quality of English Language

(x) I am not qualified to assess the quality of English in this paper
( ) English very difficult to understand/incomprehensible
( ) Extensive editing of English language required
( ) Moderate editing of English language required
( ) Minor editing of English language required
( ) English language fine. No issues detected

Yes

Can be improved

Must be improved

Not applicable

Does the introduction provide sufficient background and include all relevant references?

( )

(x)

( )

( )

Are all the cited references relevant to the research?

(x)

( )

( )

( )

Is the research design appropriate?

( )

(x)

( )

( )

Are the methods adequately described?

( )

(x)

( )

( )

Are the results clearly presented?

(x)

( )

( )

( )

Are the conclusions supported by the results?

(x)

( )

( )

( )

Comments and Suggestions for Authors

  • Firstly, thank the authors for exhaustively responding to the comments and following the various suggestions.

The purpose of the study thus appears adequately illustrated, and the manuscript improved.

Unfortunately, it seems that the authors cannot improve it with a further comparison with patients who are wild-type and so more correctly definable as "sporadic breast cancers."

This analysis would have further paid tribute to the potential of this study.

Response:

We appreciate the reviewer's acknowledgment of our efforts in addressing previous comments and improving the manuscript. We recognize the potential value of comparing patients who are wild-type using our large panel, and we will consider this suggestion for future research.

It is important to mentioned that if we had utilized a smaller multigene panel focused solely on hereditary breast cancer, our patients would have been categorized into the same groups as in our study. Moreover, each patient would have been assigned to the same group as determined in our study if we had employed a panel similar to the one provided by Invitae for Hereditary Breast Cancer Panel (# 13 genes): https://www.invitae.com/us/providers/test-catalog/test-01202

  • In light of the authors' answers, further minor comments:

-The term "sporadic breast cancers" may be misleading. Something like "breast cancer cases carrying PVs in other cancer predisposition genes" could avoid misinterpretations. Or, more simply, they could avoid the term "sporadic breast cancers" in all the text.

Response:

Thank you for your constructive feedback. We have made revisions to the terminology used in the manuscript to avoid potential misinterpretations. We added a new paragraph (highlighted in green) with the information regarding how the patients were categorized according to the genetic testing results. In accordance, these Terms are now implemented throughout the manuscript (with tracking changes in blue) and in labels on Figure 2.

Editions were added in the Results section 3.2 (page 5, paragraphs 1, 2 and 3): “In the present study, breast cancer patients were categorized into two groups based on genetic testing results. The first group, referred to as “associated gene carriers”, consists of patients carrying PVs in genes known to confer a 17% or higher absolute risk of breast cancer, as per NCCN guidelines [5]. The second group, termed “non-associated gene carriers”, comprises cases with PVs in other genes not associated with breast cancer susceptibility and who are wild-type for associated genes.

Most of the 82 breast cancer patients were classified into the first group of “associated gene carriers”, and these were diagnosed with a hereditary breast cancer syndrome (74.4%; n = 61), referred to as inherited breast cancer cases.”

“The remaining patients (21/82) were combined into the second group of “non-associated gene carriers”, none of whom were diagnosed with a hereditary breast cancer syndrome and are referred to as other breast cancer cases.”

  • - In the introduction (in the third paragraph), the authors point out that "Despite these reports, there is insufficient characterization regarding the impact of BRCA and non-BRCA germline PVs on the survival and outcomes among the population affected with hereditary breast cancer syndromes in the Columbian context." 

However, they stated that they categorized two different groups: breast cancer patients carrying germline PVs in genes with an established association with breast cancer risk (including patients with moderate penetrance non-BRCA genes, such as ATM, CHEK2, and others) and those carrying PVs in genes not known to confer increased risk for breast cancer (they defined them "sporadic breast cancer"). 

To improve consistency and clarity, the authors could add a sentence to the introduction (last paragraph) clarifying that they will not compare BRCA and non-BRCA cases (indeed, they do it between BRCA+other breast cancer predisposition genes vs. other genes).

Response:

Thank you for your suggestion. However, we consider that this addition is not necessary, since in the Figure 2, we present survival rates for both, the “associated gene carriers” and the “non-associated gene carriers” categories, as well as stratified by carriers of PVs in BRCA1, BRCA2 and HR-DDR genes (all within the “associated gene carriers” group). We also mentioned these observations in the Results section (page 8, paragraph 3) and Discussion section (page 10, paragraph 4).

  • -In the methods, the authors could provide the average follow-up duration for all patients, both for those with mutations in genes associated with breast cancer risk and for patients with sporadic breast cancer. 

Response:

Thank you for your suggestion. The calculated median follow-up time for the entire cohort was 38.1 months. For patients carrying PV in associated genes the median follow-up time was 36.7 months and for those carrying PV in non-associated genes this was 44.1 months.

Revisions have been incorporated into the Results section 3.3 (page 8, paragraph 3): “The median follow-up time for the entire cohort was calculated to be 38.1 months. Specifically, patients categorized as "associated gene carriers" had a median follow-up of 36.7 months, while for those in the "non-associated gene carrier" group, this duration was 44.1 months. Patients within the first group (i.e., inherited breast cancer cases) exhibited a 5-year DFS rate of 66.5% (95% CI: 52.9-86.0), while for those within the second group (i.e., other breast cancer cases) the 5-year DFS was 88.2% (95% CI: 74.2-100) (Figure 2B).”

Note: We acknowledge in the Limitations section that the follow-up duration in this study may not fully capture long-term outcomes, and unadjusted estimates of DFS and OS may introduce residual confusion. However, with a shorter follow-up duration, the "associated gene carriers" group showed higher recurrence rates suggesting a potentially more aggressive disease course. Conversely, “non-associated gene carriers” showed fewer recurrences over a longer follow-up, suggesting a less aggressive disease profile. We believe that these results could serve as a basis for the formulation of prospective hypothesis-driven studies aimed at generating findings that can be extrapolated to the Colombian population.

  • -The authors responded exhaustively to the comment about the interest in observing the difference between breast cancer patients carrying germline PVs in genes with an established association with breast cancer risk and breast cancer patients with uninformative results. They could add a sentence about this issue to the discussion. 

Response:

We appreciate your feedback. We have carefully reviewed your suggestion and added a few edits at the end of the Discussion section.

Revisions have been incorporated into the Results section 3.3 (page 8, paragraph 3): “Our study echoes this trend, as evidenced by our findings on DFS and OS rates among groups categorized by PVs in associated genes (i.e., BRCA1, BRCA2 and HR-DDR or inherited breast cancer cases) and those with PVs in non-associated genes (i.e., other breast cancer cases). By examining these specific categories, our study aimed to provide insight into the prognostic implications of different genetic profiles among breast cancer patients.

This study has limitations. Being an observational descriptive cohort study, the quality of information is contingent upon the precision and accuracy of data recording. The sample size is relatively small and given the descriptive nature of the design this study does not establish a formal comparison between groups. The follow-up duration may not accurately capture long-term outcomes and survival rates. Finally, since DFS and OS were estimated without adjustment for the presence of confounding variables, such as age, molecular subtype, and chemotherapy treatment, the observed results may be susceptible to residual confusion.”
